# Therapeutic Use of the Ketogenic Diet in Refractory Epilepsy: What We Know and What Still Needs to Be Learned

**DOI:** 10.3390/nu12092616

**Published:** 2020-08-27

**Authors:** Iwona Maria Zarnowska

**Affiliations:** Department of Pediatric Neurology, Medical University, Lublin, Poland Ul. Gebali 6, 20-093 Lublin, Poland; zarnowskai@gmail.com; Tel.: +48-60-9460-822

**Keywords:** ketogenic diet, refractory epilepsy, metabolic treatment, ketosis, ketone bodies, LGIT, MAD, barriers

## Abstract

Ketogenic diet (KD) has been used to treat epilepsy for 100 years. It is a high-fat, low-carbohydrate, and sufficient-protein-for-growth diet that mimics the metabolic changes occurring during starvation. Except for classic KD, its modified counterparts, including modified Atkins diet and low-glycemic-index treatment, have gained grounds to increase palatability and adherence. Strong evidence exists that the KD offers protection against seizures in difficult-to-treat epilepsy and possesses long-lasting anti-epileptic activity, improving long-term disease outcome. The KD can also provide symptomatic and disease-modifying activity in a wide range of neurodegenerative diseases. In an era of highly available new anti-seizure medications (ASMs), the challenge of refractory epilepsy has still not been solved. This metabolic therapy is increasingly considered due to unique mechanisms and turns out to be a powerful tool in the hands of a skillful team. Despite decades of extensive research to explain the mechanism of its efficacy, the precise mechanism of action is to date still largely unknown. The key feature of this successful diet is the fact that energy is derived largely from fat but not from carbohydrates. Consequently, fundamental change occurs regarding the method of energy production that causes alterations in numerous biochemical pathways, thus restoring energetic and metabolic homeostasis of the brain. There are barriers during the use of this special and individualized therapy in many clinical settings worldwide. The aim of this review is to revisit the current state of the art of therapeutic application of KD in refractory epilepsy.

## 1. Introduction

Prolonged periods of fasting have been used to treat epilepsy since ancient times and are dated back to 500 BC [1]. Fasting as a treatment continued up to the 1920s but the true origin of classic ketogenic diet is dated in 1921 when Rollin Turner Woodyatt noted that the ketones, acetone and β-hydroxybutyrate (BHB), were formed in normal fasted subjects [2] At the same time, Dr Russell Wilder proposed the use of a special diet for the treatment of epilepsy without causing the malnutrition which occurs with prolonged starvation [3]. Wilder’s colleague Dr Mynie Peterman, a pediatrician, formulated the classic ketogenic diet (CKD) [1]. Peterman showed the diet to be very successful in children, with 95% of patients having improved seizure control, with 60% of children becoming seizure free on this regimen. Use of ketogenic diet (KD) was a common practice until phenytoin was discovered in 1938. After increasing availability of anti-seizure medications (ASMs), the application of KD became less popular due to perceived unpalatability and poor adherence. This led to fewer dieticians trained in KD administration, thus, a change in the perception of its efficacy in epilepsy. In the early 90s, however, the use of KD increased in popularity due to the highly publicized, true story of successful application of the classic KD in influential patient, Charlie Abrahams and disappointingly high percentage of refractory epilepsy (~30%) worldwide, despite the development of new ASMs. Soon later, Charlie’s parents, Jim and Nancy Abrahams, created the Charlie Foundation which made information regarding the KD therapy available to other parents, medical providers and the public [4].

Since then, the classic KD and its later modifications—the modified Atkins diet (MAD) and low-glycemic-index treatment (LGIT), intended to enhance compliance—have been used to manage patients with intractable epilepsy [5,6,7,8]. Importantly, the term KD currently refers to any dietary therapy which results in a metabolic ketogenic state. All of the variants of the KD instigate shifts in body energy metabolism away from an insulin-mediated glucose dependent state towards an increased use of dietary fat and adipose stores for energy generation. Reduced glucose and increased ketones blood levels are hallmarks of biochemical changes during exposure to KDs [9,10].

This review will investigate the biochemistry of KD relevant to its efficacy; the features of available KDs; the putative mechanisms of action; its efficacy and side-effects; and barriers to the application of KDs. The individualized approach, which is increasingly widely accepted, is accentuated. Frontiers of our knowledge of the subject are also highlighted.

## 2. Biochemistry of KD

The principle of a KD, regardless of its form, is a limited carbohydrate intake (less than 60 g per day) of foods with low glycemic response. At the beginning of KD application, the blood glucose concentration drops and stabilizes, thereby preventing postprandial insulin release (as dietary portions tent to be evenly applied during the day), resulting in the body entering a catabolic state. As glycogen stores are depleted, the body is forced into endogenous glucose production especially in the liver (gluconeogenesis) using the amino acids alanine and glutamine, lactic acid, and glycerol. When the insulin/glucagon ratio is further decreasing during KD intake, and the gluconeogenesis is not able to keep up with the metabolic demands, free fatty acids (FFAs) are mobilized from fat tissues and dietary sources in order to replace glucose as a primary source of energy. As glucose is not readily available, the brain can use ketone bodies generated by FFAs oxidation in the liver as alternative major source for aerobic energy production [11,12]. Although the brain, unlike any other organ in the body, has an absolute minimum requirement of glucose, when ketone bodies concentration in the blood attain the range of 2–4 mM, these can satisfy as much as 60% of the brain’s energy requirements [13]. Continuously low insulin levels lead to increased uptake of FFAs into the mitochondria of liver cells via carnitine palmitoyltransferase (CPT-1). Once entering the mitochondria, fatty acids are broken down into acetyl-CoA via β-oxidation. During the classic KD, the liver oxidizes FFAs at high rates resulting in an overproduction of acetyl-CoA. Surplus acetyl-CoA is shunted to ketone bodies (KBs) production via ketogenesis. A low insulin level triggers enzymatic conversion of two acetyl-CoA molecules to acetoacetate (ACA) that can be spontaneously degraded to acetone or metabolically converted to β-hydroxybutyrate (BHB). All these three so-called KBs build up in the blood during prolonged exposure to KD and can be delivered to the brain via monocarboxylate transporter 1 (MCT-1). The KBs are interrelated: BHB and ACA can convert into each other, while ACA can turn into acetone. Extra ketones are eliminated through urination, or in case of acetone, through breathing. Once ACA and BHB reach the brain, KBs are metabolically converted into acetyl-CoA that enters the tricarboxylic acid (TCA) cycle. Then, the reduced coenzymes NADH and FADH2 allow ATP synthesis via oxidative phosphorylation by the electron transport chain in the inner mitochondrial membrane. Importantly, KBs produce more ATP in comparison to glucose, allowing the efficient maintenance of fuel production even during caloric restriction [11]. Ketogenesis in the liver is a physiological adaptation to starvation or a very low-carbohydrate diet resulting in plasma ketones of 2–5 mM after a week or so, and has never been shown to alter acid-base balance or to lower blood pH [14]. Continuously increased plasma levels of KBs and lipids along with low and stable level of serum glucose and normal blood gas create the so-called state of “nutritional ketosis” provoked by KDs [15,16]. It is essential to distinguish between nutritional ketosis and diabetic ketoacidosis, which is a potentially life-threatening complication of uncontrolled diabetes mellitus, occurring in the setting of hyperglycemia with relative or absolute insulin deficiency, the profound metabolic acidosis with high concentration of ketones and subsequent drop in blood pH [17]. Ketosis appears to be a natural adaptive mechanism which has been instrumental in the biological survival of humanity and, perhaps, essential for the evolution of the human brain [13,18]. The stable nutritional ketosis provides a steady fuel source for the energy-intensive tissues such as brain and muscle, preventing the likelihood of disruption in energy availability [19]. Of note, available clinical data have failed to show any constant correlation of ketonuria (ACA level in urine), ketonemia (BHB blood levels) or breath acetone and seizure control [20,21,22]. KB level in the blood and/or urine serves clinically as an easy and relatively cost-effective indicator of KD adherence [23]. Despite extensive research, it is still unknown how/why a high-fat diet is beneficial for a neuronal disorder such as epilepsy; however, one possible explanation is that KBs are a major fuel source for brain development in utero and in infancy. Due to high energy requirements for early brain development the most effective and easiest way of energy use is preferred [24]. Very high energy demands are required in tissue development and repair such as in epilepsy, thus, re-implementation of KB-based brain metabolism may establish an environment allowing vulnerable brain networks to repair structural or functional damage more efficiently [25]. Other measurable parameters such as quality and quantity of FFA or optimal glucose levels and their correlation to seizure control have not been investigated in depth. If such a correlation does exist at all, it could be used to reach optimal seizure control or possibly even tailor the KD to the individual patient. Nevertheless, in the high-tech era, the best marker of the KD efficacy remains the clinical history [21] (Figure 1 and Figure 2).

## 3. Available Ketogenic Diets

Current evidence suggests that only strict long-term adherence to the KD may give rise to fundamental shift from glucose-based metabolism to nutritional ketosis which is both necessary and sufficient for its clinical efficacy for medication-refractory epilepsy [10]. To introduce and maintain nutritional ketosis, KDs must be restricted in carbohydrates to some extent, enriched in fat and should provide adequate protein to ensure optimal growth and/or protect lean body mass which is especially important in children. The decision to use KDs in refractory epilepsy is dependent on the availability of an experienced and dedicated keto-team, therapy’s appropriateness to patients’ medical condition, patients’ motivation and ability of their families/caregivers to comply with the diet. Unlike any anti-seizure drug therapy for epilepsy, the KD is a multidisciplinary therapy which requires the knowledge, support, and partnership cooperation of the patient’s epileptologist, nurses, dietitians, social workers and the “keto-family” [26].

Traditionally reserved as a “last treatment option” in pharmaco-resistant epilepsy in childhood, currently KD should be considered earlier for difficult-to-manage epilepsy in patients of all ages and seizure types in variety of settings [21,23]. The pre-KD assessment and pre-KD preparation process consists of setting and agreeing on realistic goals for dietary therapy and clinical outcomes between the keto-team and the patient/families, and is key for KD success. Families/caregivers must understand their pivotal role in KD therapy, including time devoted to meal preparation, costs of foods and supplements, and potential side-effects or behavioral changes in the child that will challenge KD therapy. The KD requires commitment for the recommended trial period of at least 3 months to determine its efficacy [23,27]. Due to fundamental changes of intermediary metabolism provoked by KD and the inherent danger of side-effects, all patients on KD should be seen for routine clinical follow-up visits (1, 3, 6, 12, 18 and 24 months after diet initiation) [23]. The KD for epilepsy, if successful (seizure freedom or ≥50% seizure reduction), is traditionally continued for 2 years. However, the origins of this tradition are unknown [23,28], therefore controlled trials are needed to confirm that timeframe. Children with pyruvate carboxylase deficiency, fatty acid oxidation defects or any other syndromes in which energy metabolism is compromised could experience metabolic decompensation during dietary changes or periods of illness, therefore, all children considered for KD must undergo a urine and plasma metabolic screening as well as serum lactate level prior to KD administration. As baseline acidosis might worsen with the KD, a baseline blood gas analysis is required [23,29].

Despite numerous attempts to reduce the restrictiveness and prescriptiveness of the CKD, it remains a challenging task for both the patients and/or caregivers and the keto-team. Therefore, alternative ways generating a nutritional ketosis state and therefore the diet administration have been developed in the last 20 years. Modified KDs changed the prescription rules (calories, ratio of fat to carbohydrate, quality and quantity of macronutrients content, methods of diet implementation and maintenance) allowing easier maintenance, palatability, and tolerability thus improving adherence to KD therapy.

There are currently four main forms of KD available: 1. The classic KD (CKD), 2. The medium chain triglyceride ketogenic diet (MCT KD), 3. MAD, and 4. LGIT [23].

The classic KD (CKD) is perceived as the most restrictive and “prescriptive” form of KD [30]. Prescription requires all foods and beverages to be calculated based on “ketogenic ratio”: the proportion of fat to protein plus carbohydrate combined by weight in grams. The goal of the CKD is to achieve the pre-defined ratio of fat to carbohydrate and protein (in terms of grams). Energy requirements and daily protein intake must be pre-defined, as it forms the basis of an individual carbohydrate and fat calculation to perform the dietary prescription. Once the macronutrients in grams are established, the amounts are divided up evenly into meals and snacks, so the ketogenic ratio is the same in every meal over the day. It is required that foods and beverages be calculated and precisely weighed on a gram scale. The traditional goal of CKD prescription is 4:1 ketogenic ratio, or 3:1 (for infants, adolescents, or those for whom higher protein or carbohydrate load is desired). In the 4:1 CKD fats provide 90% of dietary energy, mainly in form of long-chain triglycerides (LCTs), which requires carnitine for hepatic metabolism. In the context of cardiovascular and metabolic health, inclusion of balanced mixture of saturated fat, mono or/and polyunsaturated FFA is prudent. Interestingly, lower ratios as 2:1 or 1:1 may be as efficacious as higher ratios for seizure reduction on individual patient basis. The traditional method of initiating the CKD involves a hospital admission with a period of fasting (12–48 h), protein restriction to 1 g/kg of body weight, fluids restriction to 60–75 milliliter/kg of body weight and later reduced dietary calorie level to 80–90% of the estimated daily requirements [31]. Inpatient initiation allows occasional metabolic fluctuations (symptomatic hypoglycemia, severe acidosis, dehydration) and gastrointestinal system complications to be closely monitored and corrected promptly. As the CKD is available as formula preparation, it can be considered to be treatment option in patients fed via gastrostomy tubes or in patients in intensive care unit settings [32,33,34]. Seizure control on CKD evolves within days to weeks of starting the diet [21,28]; however, serum or urine KBs level does not correlate tightly with clinical effects [35,36]. Therefore, ketosis cannot be used as a marker for CKD clinical efficacy, moreover, rapid increase in blood glucose can reverse the therapeutic effect within minutes [37]. The CKD being particularly stringent, with high amount of saturated fats (butter, cream, mayonnaise, animal fat) and cholesterol, yet offers highest ketogenic potential among other KDs [21]. Concurrently, 100 years of clinical experience indicate that CKD is not a benign therapy, as it is associated with several short-term complications during the initiation phase and adverse effects during its maintenance [38]. The most common side-effects are preventable and potentially treatable, and are rarely a cause of CKD discontinuation [39]. In a retrospective analysis of 32 children with Dravet syndrome treated with CKD, none of the patients had side-effects severe enough to require withdrawal of the diet. Furthermore, it was found that noncompliance was more frequent in solid-fed older children compared with infants fed with the CKD as a liquid formula [40]. The restrictive character of the CKD is considered one of the main reasons for discontinuation. In a series of 150 children and adolescents from Johns Hopkins KD center, 17% discontinued the CKD during the first 3 months. At 1 year, 45% of patients who had started the diet had discontinued it due to either insufficient clinical effects which did not meet expectations, or due to its restrictiveness [39]. Based on clinical studies from the last 20 years, most KD centers around the world are more flexible in the implementation of the CKD. Presently, CKD can be started on an outpatient basis, without fasting period or fluid restriction, using a “low ratio slow initiation method” [21,41,42]. This approach, reducing both restrictiveness and “prescriptiveness” of the traditional CKD protocol, increases adherence to CKD.

The medium chain triglyceride ketogenic diet (MCT KD) is a variation of CKD in which the proportion of LCT fats is replaced by MCT oils comprising about 60% triglycerides of caprylic acid (CA8) and 40% of capric acid (CA10) [43]. The MCT KD was introduced by Dr Huttenlocher of the University of Chicago, as he noticed that MCT are absorbed more efficiently than long-chain fat, are carried directly to the liver in the portal blood, and do not require carnitine for it transport into cell mitochondria for oxidation [44]. By contrast to LCTs, MCTs can cross the blood-brain barrier and affect brain metabolism directly, providing an alternative energy source for neurons and astrocytes [43,45,46]. Additionally, CA10 have been shown to have a direct inhibitory role in seizure control via selective inhibition of AMPA receptors in animal seizure models [47]. Just as CKD, MCT KD requires thorough calculation by dieticians. Prescription is not based on diet ratio but requires calculation of the percentage of total daily energy from LCT, MCT, protein, and carbohydrates. These can then be converted in grams of MCT oil and food choices and distributed into daily meal plans. As MCT oils produce higher levels of ketosis than LCTs, less total fat is needed in the MCT diet. Some 45–50% of energy from MCT is likely to achieve gastrointestinal tolerance and level of ketosis comparable with CKD, but can range from 30–70% based on an individual patient’s tolerance. MCT oil can also be used as a supplement to CKD meals to boost ketosis or to treat hyperlipidemia and constipation attributable to CKD [23]. MCT KD has similar efficacy to the CKD, but in comparison permits proportionally less total fat and more carbohydrate and/or protein to be included making it more palatable especially for children with limited food options [35,48].

The MAD was first developed in the early 2000s in the John Hopkins Hospital (Baltimore, USA) as a result of the observation of seizure reduction in patients using the low-carbohydrate and high-protein diet originally intended for weight reduction [6]. The MAD was designed to offer greater flexibility with food choices and fluids as well as reduced involvement of a keto-team in pre-diet preparation, initiation, and monitoring. It is perceived as less restrictive and more palatable in comparison to CKD and may be associated with a greater compliance by older children, teenagers, and adults. This type of KD is usually started without a fast and on an outpatient basis. Pre-diet teaching and dietitian support is limited to 1–2 h of how to count food carbohydrate gram, and might be conducted even by email correspondence [49]. Food options on MAD are usually provided by food exchange lists. It allows patients to be more independent in terms of using premade products or eating out. Energy requirements are not typically assessed beforehand; however, some protocols recommend calories to be restricted to 75% of recommended daily intake to boost ketosis [50]. The daily carbohydrate consumption is very restricted, but it can be consumed at any time with fats. An initial goal of grams of carbohydrates is set for 10 g/day for children and 15–20 g/day for adolescents and adults in the first weeks, but it can be modified with an interval of at least 1 month depending on seizure control up to 25–30 g/day on an individual patient basis. Protein is not overly restricted but may need regulation to promote ketosis. High-fat foods intake is encouraged to satisfy appetite and meet individual energy requirements [51]. Once attained, nutritional ketosis on MAD can be controlled at patient’s/families own discretion measuring ACA level in urine by using commercially available reagent sticks on weekly basis [51]. 

In randomized controlled trials, MAD was found to be significantly more effective in controlling seizures in children and adults with refractory epilepsy, as compared to the continuation of anticonvulsant drugs alone [52,53,54]. Cochrane’s review compared efficacy of ketogenic therapies for epilepsy, based on 11 randomized controlled trials, comprising 712 children and adolescents and 66 adults. Seizure freedom was reported in up to 25% of children and adolescents in MAD group after 3 months; ≥50% seizure reduction in up to 60% of children and adolescents. In comparison with CKD 4:1 group where seizure freedom was reported in up to 55% of children after 3 months and ≥50% seizure reduction in up to 85% of patients. In MAD group for adults, ≥50% seizure reduction was reported in 35% of patients. No seizure freedom was reported in adults on MAD. The 4:1 CKD was consistently associated with more adverse effects compared to MAD [50,55]. Despite the higher efficacy of CKD than MAD, MAD is much better suited for adolescents and adults allowing flexibility, independence on home lodging, less keto-team involvement, and the possibility to use it as weight regulator. Consequently, adherence in this age group is much better for MAD than CKD. Due to slower implementation of MAD than CKD, many of the metabolic fluctuations experienced during the initial few weeks of MAD could be avoided [56,57]. When CKD causes too many side-effects or is found to be too restrictive, it could be transformed into MAD resulting in more favorable clinical profile [58]. Additionally, it was found that MAD is helpful for children with epilepsy and behavioral problems in which KD’s restrictiveness would be challenging [59]. The question arises, if MAD could be a treatment of choice for children with autism spectrum disorder (ASD) and epilepsy even before ASMs [58,59,60].

LGIT has been developed in Boston, USA in 2002, based on the observation that rapid changes in blood glucose and insulin levels may provoke seizures in patients with epilepsy treated with CKD. The goal of LGIT is to maintain relatively stable blood glucose levels and to prevent large postprandial insulin fluctuations [8,61]. LGIT is a less restrictive diet in terms of food options compared to CKD, yet more “prescriptive” than MAD. Nevertheless, restrictiveness has been reported as main reason for the discontinuation of LGIT (24%) [22]. Initiation of the LGIT is usually established in an outpatient setting. Pre-defined carbohydrate content is set at approximately 10% of daily energy requirements, which is estimated based on food records completed prior to diet initiation. It reflects the physiological maximum below which the body will use fat as primary energy source (40–60 g of carbohydrate per day, including fiber). Only carbohydrate foods with glycemic indices of less than 50, relative to glucose, are allowed. Glycemic index (GI) represents the tendency of foods to increase blood glucose after feeding, indexed to ingested glucose = 100 [8]. Carbohydrates are evenly distributed and consumed with fat and protein. Protein content is based on approximately 30% of energy prescription, provided as an average (estimated) portion of protein for a meal. Fat prescription is based on portion guidance (approximately 60% of target energy) to ensure adequacy. LGIT is lower in fat and higher in carbohydrates, yet similar efficacy to the traditional CKD [8]. Intriguingly, clinical efficacy of the diet seems to correlate with more stable blood glucose and probably insulin levels and do not correlate with ketosis, which is inconsistent in LGIT [22]. Even if the concept of LGIT is relatively new to the field of neurology, diets with low GI are linked with favorable health measures for diabetes, heart disease, obesity and polycystic ovary syndrome [62,63], perhaps it is needed to conduct studies with epilepsy and such comorbidities. 

A modified ketogenic diet (MKD) is a hybrid KD, adopting principles from other established KD protocols and defining new elements to the MKD. It was designed to find optimal balance between ketosis and palatability in particular patients. In fact, presently the majority of KDs are individualized MKDs, differing from each other, tailored to individual patients’ needs (energy requirements, macronutrients allowance) in order to find optimal equilibrium between efficacy and adherence. Modifying the restrictiveness of CKD can be helpful when starting the diet or when tapering down to a more sustainable long-term diet. Macronutrients are adjusted over time from 3:1 to 1:1, decisions are based on ketone levels and seizure reduction rate in most cases [64]. MKD can be effective in adults, although, even with regular nutritional support, retention rates remain low [65].

Recently, a healthy and more palatable diet with effective anti-epileptic properties by combining different nutritional strategies was designed by Dallerac and coworkers [66]. This new combined diet (CD) comprises MCT, polyunsaturated fatty acids (PUFA), LGIT and a high proportion of branched-chained amino acids (BCAA) vs aromatic amino acids (AAA). Interestingly, BCAA have been proposed to favor ketosis and GABA synthesis while reducing glutamate levels, thus, decreasing brain excitability. Efficacy of this diet has been confirmed only in rodents but in theory this approach seems to be promising especially due to low side-effect profile [66].

Unique features of KD and metabolic therapies that differentiate them from ASMs are:The effects of KDs do not significantly impair current brain function (no influence on synaptic function, input–output excitability or paired impulse, stimulation-induced plasticity and very little on long-term potentiation [67,68]);The effects of KDs are not only non-neurotoxic but also neuroprotective and disease-modifying. In contrast to ASMs that exert only anti-seizure activity, KDs may have anti-epileptic and disease-modifying properties [69];The effects of KDs may last much longer than application of drug treatments themselves; shift away from seizure generation may take some time to occur [70];“Reverse dilemma” compared to most drugs used in clinical practice. With the ASMs, usually, a disease or pathology exists, and we seek a therapy based on the pathophysiology of the disease and molecular mechanism. With the KD, we have the opposite situation—the therapy is effective, but our goal is to understand the mechanism of action, which is largely obscure. This situation is favorable though, at least in terms that KD does work without knowing how it works.

Recent clinical experience teaches us that despite tremendous efforts on development of new ASMs, with distinct mechanistic targets, the number of patients with refractory epilepsy is not declining. There is a need to consider different treatment options and not necessarily be concentrated on new discoveries of ASMs. Bearing in mind unpalatability of the KD, its relative complexity leading to adherence problems, there is a need to introduce simplified versions of the diet, tailored to the individual patient needs. It is important to consider that key mechanisms for the clinical efficacy may differ for each patient and depends on the seizure condition itself (e.g., the level of ketosis might be critical for the KD clinical efficacy in infantile spasm, whereas it might be calorie restriction for Doose syndrome [58]), age of the patient and may vary based on the clinical protocol or duration of diet administration. If so, the question remains open, if the KD can be packaged into a pill.

## 4. Putative Mechanisms of Action

Over the past two decades along with an explosion in clinical use of the KDs, research studies have been exploring the fundamental question of how the KD works. In fact, there is a myriad of biochemical and physiological changes evoked by consumption of dietary treatments that can be potentially therapeutic for epilepsy. It is apparent that the shift in energy generation from carbohydrate sources to lipids, essentially alters many fundamental biological systems and both molecular end products and intermediates in virtually all metabolic pathways. However, it is not straightforward to determine cause-effect relationship because one cannot be sure whether specific molecular and cellular alterations are relevant or simply represent epiphenomena. Enquiring research confirming the previously known mechanisms of KD usually end up providing new possible anti-seizure mechanisms. Moreover, evidence suggests that effects of KD may either occur rapidly or develop over time. This diversity of mechanisms is increasingly appreciated as a major strength. As the epileptic episode arises from largely unknown, complex, probabilistic, and occult set of variables and occurs in a susceptible and unstable neural network, it is plausible to assume that novel, unique, and multiple mechanisms offered by the KD may be extremely helpful, nevertheless, it is rational not to expect absolute efficacy [10].

Taken together, these mechanisms can be grouped into changes that alter neurotransmission at synapses and those that improve neuronal and glial structure and homeostasis; although, there are many overlapping processes.

Enhanced bioenergetic effect. Ketones may provide a more efficient source of energy for the brain than glucose. Microarray studies showed that ketones induced coordinated up-regulation of genes encoding energy metabolism and mitochondrial enzymes, increasing the number of mitochondria in neurons and glia [71,72]. Better energetics is proposed to limit seizure activity by stabilizing neuron resting potential, so they hyperpolarize or activate K_ATP_ channels through adenosine release [73,74]. In KD the reduction in the use of glucose is additionally enhanced by inhibition of glycolysis by shifts in the mitochondrial ratios of acetyl-CoA/CoA and NADH/NAD^+^ [75]. The increase of ATP generation may either provide cellular energy or influence ion channels function, neurotransmitters, and transporters. The end results are increases of the energy reserves of neurons, improving neuronal homeostasis and higher resilience to neural damage occurring during seizure episodes.

Some human and animal studies support the anti-seizure mechanism by showing that KD leads to increased GABA and reduced glutamate in the brain. Another potential player is the inhibitory neurotransmitter neuropeptide Y (NPY) known to be anti-epileptic but up-regulated in ketogenesis [10].

Action of ketone bodies. Elevated blood KB levels represent the most consistent metabolic biomarker of adherence to the KD. It was documented that direct injection of acetoacetate and acetone in animal models may directly suppress seizure activity [76]. However, other studies showed that ketone levels may not correlate with seizure control [77]. Recent data support the growing recognition that KBs are not only substrates for energy production but also possess pleiotropic mechanistic properties that altogether exert a net anti-seizure effect [78,79]. Indeed, β-hydroxybutyrate has been documented to interact with multiple novel molecular targets such as the mitochondrial permeability transition pore, histone deacylases, hydroxycarboxylic acid receptors on immune cells and NLRP3 inflammasome [79].

Role of glucose. Classic observation of reduced glucose levels in patients successfully treated with the KD is accompanied by the experimental evidence that low or stable glucose levels reduce seizure susceptibility and inversely, increased glucose levels can precipitate seizures [37]. LGIT can be effective in reducing seizures without producing detectable ketosis [22]. Therefore, it is plausible to assume that low and stable levels of glucose might be a key mechanism of KD. Recently, the most encouraging evidence of this paradigm is the current preclinical development of 2-deoxyglucose (2-DG), a glycolytic inhibitor [80]. If the effectiveness of the KD can be enhanced by adjunctive therapy with agents such as 2-DG or antidiabetic drugs such as metformin, is still unknown.

Antioxidative and anti-inflammatory effects. Mitochondrial respiration, while generating ATP, also produces many reactive oxygen species (ROS). KD can induce up-regulation of mitochondrial uncoupling proteins which correlates with decreased ROS generation and increased resistance towards seizures. KD also increases the body’s own antioxidant defense system, namely glutathione levels, and protects mitochondrial DNA from ROS damage. The end result is improved resilience of brain cells to the damagingly high levels of ROS that are generated in a seizure [81,82].

Local changes in pH. Ketone metabolism generate pH-lowering metabolites, hence, a change in pH was proposed as a mechanism how KD influences brain function. However, there is no evidence that KD significantly lowers brain pH, although mild decreases in pH may be possible in local microdomains. Indeed, local compartments have been shown to exhibit differential pH regulation during neuronal activity. Many receptors are modulated by pH, such as acid-sensing ion channel 1a (ASIC1a), NMDA receptors and GABA receptor isoforms, involved in seizure generation and epilepsy [83,84,85].

Histone deacylases—epigenetic effect. ΒHB is an inhibitor of class I histone deacylases. They are epigenome modifiers that by removing acetyl groups from histone tails, compact chromatine. Consistent with resulting increased enzyme activity, treatment of mice with BHB conferred substantial protection against oxidative stress [86].

Recent evidence suggests that the kynurenine pathway is involved in the neuroprotective and anticonvulsant activity of the KD [87,88,89]. This pathway generates a range of metabolites collectively known as kynurenines, which are involved in a variety of medical conditions such as inflammation, immune response, and several central nervous system (CNS) disorders including epilepsy, depression, and diseases associated with neurodegeneration [90,91,92]. Recent reports the possible role of kynurenic acid, a putative endogenous neuroprotectant and anti-epileptic agent, in the mechanism of efficacy of KD [88]. Glutamate decreases the production of kynurenic acid in bovine retinal slices, the effect is attenuated by KBs overproduced during KD. It was also found that chronic exposure to KD differentially increases concentration of kynurenic acid in discrete brain structures of young and adult rats [93]. Recently, we reported a pattern of changes in the blood levels of kynurenines or its ratios in patients with refractory epilepsy after starting the KD. Notably, higher concentration of kynurenic acid and lower concentration of kynurenine were found in patients who attained a higher reduction in seizure frequencies on KD (*p* < 0.05) [94]. However, whether specific kynurenines at pre-KD baseline or when on the KD could serve as predictive or prognostic biomarkers in refractory epilepsy warrants further investigation.

Microbiome alterations. The KD alters gut microbiome in mice and humans [95,96]. Recently, it was shown that KD-induced microbiome enrichment in *Akkermansia* and *Parabacteroides* species restores seizure protection in two mouse model of epilepsy, moreover, microbiome transplantation confers seizure protection in the same models in mice fed control diet. It is suggested that these effects correlate with alterations in hippocampal metabolomics profile including elevated hippocampal GABA/glutamate levels [97]. Human studies are lacking; however, Xie and coworkers reported that gut microbiome patterns in healthy infants differs dramatically from that of the epileptic group. KD could significantly modify symptoms of epilepsy and reshape the gut microbiome of epileptic infants [98]. As causal relationship is not proven, precise gut-brain link leading to anti-epileptic efficacy of KD warrants further investigations.

## 5. Efficacy of the KD

Presently, KD can effectively treat epilepsy in individuals from infancy through adulthood [23]. Originally, adolescents and adults typically had not been considered candidates for KD; however, with time some research reported similar outcomes in adults [99]. In fact, there is evidence that children younger than 2 years tend to respond to the KD treatment better. Patients in this age group are in the process of rapid brain maturation, prone to be affected by serious epileptic encephalopathy. If seizures are not controlled rapidly brain damage would be irreversible [40,50]. Potential patients could be divided into several subgroups.

At present, KD is mainly used in refractory epilepsy, i.e., in patients in whom two ASMs turned out to be ineffective [4]. However, there are some epilepsy syndromes and conditions for which KD has been reported as more beneficial (>70%) than the average 50% KD response (defined as >50% seizure reduction) e.g., Dravet syndrome, infantile spasm, tuberous sclerosis complex. Vast majority of the consensus group [23] believe that KDs should be strongly recommended very early during treatment of these syndromes. Several conditions in which KD has been reported as moderately beneficial (not better than the average dietary response) e.g., childhood absence epilepsy, cortical malformations, juvenile myoclonic epilepsy, Rett syndrome, Landau–Kleffner syndrome and Lennox–Gastaut syndrome (LGS) [23].

Primary therapy of first choice for specific conditions, e.g., Glucose transporter type 1 deficiency syndrome (GLUT1 DS), pyruvate dehydrogenase complex deficiency (PHDH) syndrome. In both disorders, the concept of KD is to provide ketones that bypass the metabolic defects and serve as an alternative cerebral fuel. Additionally, neuroprotection offered by a KD might reduce disease severity [100,101,102]. In these patients KD therapy should be used instantaneously after diagnosis before any pharmacological treatment has been implemented and should be maintained into puberty [23].

Some patients choose KD as a primary treatment before application of ASMs (some patients want to avoid pharmacological treatment, if KD is successful they do not go for ASMs).

Some patients should avoid KD even as a secondary treatment as it may delay more efficacious surgical therapy of epilepsy. There is evidence that children with epilepsy due to a focal lesion may do less well with KD than with respective, excisional surgery [103].

The first large prospective study assessed the efficacy and tolerability of the KD on 150 consecutive children for a minimum of 1 year at John Hopkins Hospital. The authors reported at least 50% seizure reduction in 71%, 73% and 90% of patients, at 3, 6, and 12 months, respectively regarding the patients that remained on the diet, success rate was 59%, 50 % and 50%, respectively, regarding the patients that initiated the diet. This study did not demonstrate greater efficacy in any particular seizure type (i.e., myoclonic/drops vs. tonic-clonic) although epilepsy syndromes were not delineated specifically [39]. Soon after, the authors determined that KD is very effective in decreasing the incidence of atonic or myoclonic seizures in children with (LGS) even within the first 48 h [104].

Three to six years after initiation, the KD had proven to be effective in the control of difficult-to-control seizures in children (27% seizure free or almost seizure free). The diet often allows decrease or discontinuation of medication. They reported that at least 50% of patients benefited from the reduction of at least one anticonvulsant drug. Overall, they found it is more effective than many of the newer anticonvulsants and is well-tolerated when it is effective [39,105].

Neal et al., reported in a prospective randomized study the clinical efficacy of KD vs control group at 3 months [27]. Overall, a reduction of 75% in seizure frequency was observed. Twenty-eight children (38%) in the diet group had greater than 50% seizure reduction compared with four (6%) controls (*p* < 0.0001), and five children (7%) in the diet group had greater than 90% seizure reduction compared with no controls (*p* = 0.0582) [27].

Consistently, today’s clinical investigators have reported that ~50% of patients with medically intractable epilepsy experience a ≥50% reduction in seizures while on KD, and 10–20% of such patients—typically children—can maintain seizure freedom even after weaning from anti-seizure drugs and dietary restrictions [9,10,106,107].

Kossoff et al., evaluated MAD efficacy and safety and reported a 50% seizure reduction in 50% of patients with over 90% reduction in 28% of patients [7]. A relatively recent paper comparing CKD and MAD shows 50–60% reduction of seizures with no significant differences; however rate of seizure freedom was 53% for CKD and only 20% for MAD [50].

The use of LGIT initially reported by researchers from Massachusetts General Hospital [8] seemed to be encouraging. Later, authors reported at least 50% reduction in 50, 54 and 66 % of patients at 3, 6, and 12 months, respectively with relatively limited side-effects. Over 90% seizure reduction rate occurred in 45% patients. Of note, the most cited reason for the discontinuation of the diet was its restrictiveness (24%) [22]. First Italian experience shows beneficial effect of LGIT on some patients (multifocal epileptic encephalopathy, Lennox–Gastaut syndrome, Dravet syndrome) but the results were not as good as reported by Pfeiffer and Muzykewicz [8,22,108].

Own experience shows ≥50% reduction in seizure frequency in 85% of patients, with very careful, individualized dietary approach with relatively even allocation to either classic KD or MAD in relatively small group of 16 patients. The method used consisted of application of least cumbersome type of diet with concomitant maximal effectiveness, which is possible to achieve with very good cooperating patients and strict physician’s supervision [94].

Recently, Cochrane Database Search of the results of randomized or quasi-randomized controlled trials of KDs for individuals with drug-resistant epilepsy was analyzed. Altogether, the 11 studies recruited 778 patients; 712 children and 66 adults. Reported rates of seizure freedom reached as high as 55% in classical 4:1 KD group after 3 months and >50% seizure reduction might occur up to 85% of children. Assessment of seizure freedom after MAD treatment showed 25% seizure freedom and reduction rate up to 60% after 3 months. Although there was some evidence for greater anti-epileptic efficacy for classical 4:1 KD over lower ratios, the classical KD was consistently associated with more adverse effects [55].

Having in mind neuroprotective and disease-modifying properties of KDs, it should be stressed that their efficacy is not only limited to seizure control [69,109] but also KD effects on neurobehavioral development, cognitive functions, and sleep quality. The neurobehavioral improvements comprise adaptability, gross motor movements, fine motor movements, language and social interactions [110,111]. Cognitive benefits comprise alertness, attention, and global cognition. There are indications that these improvements are caused by both seizure reduction and direct effect of KD on cognition [112,113]. In addition, KD decreases sleep and improves sleep quality by increasing REM sleep [114]. All these factors contribute to the improvement of quality of life.

Assessment of KDs efficacy is concentrated on the effect of diets on seizure control understood as reduced fits count. However, number of studies about the influence of KD on disabilities related to epilepsy itself is limited and should be increased. Even in patients in whom seizure reduction is not dramatic, the improvement of quality of life is frequently observed together with the reduction of the number of ASMs [115].

ASMs dosage adjustment might be some concern while applying KD. In a prospective study, it was found that MAD reduces the plasma concentration of ASMs (clobazam, carbamazepine, valproate, lacosamide, lamotrigine and topiramate) with a mean of −10%. In addition, it was negatively correlated with urine ketosis [116]. However, other authors did not observe pharmacokinetic interaction while initiating the diet. Thus, dosage adjustment does not seem to be justified [117,118].

## 6. Tolerability and Adverse Effects

Effectiveness and tolerability of anti-epileptic treatment are inextricably entwined. In fact, ineffective KD is poorly tolerated even, if devoid of significant side-effects, leading to discontinuation of the treatment. Thus, tolerability or retention time on treatment becomes a surrogate for efficacy and side-effects. Looking at the efficacy of KD against decreasing number of patients, it is almost obvious that drop-out refers to the patients that do not achieve expected success [22,39,119,120]. Freeman (1998) reported 42% drop-out rate during the first 12 months of the CKD treatment, among children and adolescents. Thirty-five percent of children and adolescents discontinued due to poor efficacy and 7% remaining due to other reasons taken together [39]. Later, in the study from the same diet center for epilepsy, long-term follow-up of 87 MAD-treated children was reported. Despite the fact that the adverse effects were mild (predominantly elevations in lipid profile and gastrointestinal upset) and that MAD is less restrictive in comparison with CKD, only 55% of children treated with the MAD continued diet beyond 6 months [119].

Short-term adverse effects of the KD occur commonly during the initial few weeks of dietary therapy. The most common ones involve risk of hypoglycemia, lethargy, irritability, metabolic acidosis, vomiting, dehydration, diarrhea and refusal to eat [38]. As these complications are secondary to adaptation to metabolic shift induced by KD, most of it appear very predictable and potentially preventable, so it does not have to be an obstacle to diet continuation. As hypoglycemia, acidosis, and dehydration are more frequent with traditional, rapid initiation of KD [21,121], diets are increasingly initiated outside of the traditional manner (e.g., in an outpatient setting, with no fasting, no fluid or calorie restriction, with a liquid formulation, with gradual, slow initiation method). Pre-diet counseling process when patients/caregivers as well as patient’s GP, are instructed of how to avoid anticipated adverse effects and how to manage once they occur, might significantly increase adherence to KDs [23,122,123].

The gastrointestinal system disturbances are consistently noted during short-and long-term KD therapy. Constipation, abdominal pain, emesis, gastroesophageal reflux disease is experienced by up to 50% of children during KD. Most gastrointestinal complaints could be alleviated by fluid intake adjustments, dietary manipulation, and laxatives [23].

The long-term side-effects of the KD are routinely monitored during scheduled clinical follow-up. At each stage of the KD, the risk/benefit ratio regarding a patient remaining on the KD needs to be assessed [124]. Supplementation of vitamins, minerals, trace elements, if tailored to the patient needs and taken regularly, may lead to avoidance of many of these side-effects.

According to the still-prevalent lipid heart hypothesis, high dietary fat intake while on KD, causes poor cholesterol profiles and elevated cardiovascular risk especially in adult populations [125]. An increase in plasma lipids in patients on KDs can be an immediate as well as long-term manifestation of nutritional ketosis, and is a well-known “side-effect” of almost all currently used KDs [126,127,128]. Standard lipid panel, which is commonly ordered during clinical follow-up visits, report increased levels of serum triglycerides, total cholesterol, low-density lipoprotein cholesterol (LDL-C) in up to 59% of children who consume CKD [128,129]. However, serum lipid profile alterations seem to be temporary and normalize by 12 months of the CKD [126,130,131]. Moreover, simple dietary adjustments including increased consumption of flaxseed oil or olive oil (which contain mixture of monounsaturated fatty acids, saturated fatty acids and both n-3 and n-6 polyunsaturated fatty acids) in exchange of saturated fat, supplementing with MCT oil or L-carnitine, decreasing the KD ratio may prevent KD-induced hyperlipidemia [23,132]. Additionally, growing evidence strongly suggests that in addition to standard lipid profile, analyses of lipid sub-fractionations may be required to identify patients at high cardiovascular risk, and to inform optimal clinical recommendations [133,134].

Subjects treated with KD may have higher arterial stiffness parameters (early markers of the vascular damage) and increased levels of cholesterol and triglycerides compared to normal diet patients [135]. These elements may contribute to the development of atherosclerosis later in life. Other authors confirmed unfavorable influence of KD on lipid profile but not on arterial elastic properties [126,136].

Nephrolithiasis has been reported in approximately 6% of children on the CKD (Furth et al., 2000). Renal calculi formation (most commonly uric acid or calcium oxalate stones) might be attributed to the metabolic effects of the KD, particularly uric acidemia, hypocitruria, hypercalciuria, and aciduria. Children who are non-ambulant, with a high calcium/creatinine ratio before starting the diet, and those maintained on KDs for more than 2 years being those, who are at risk the most [137,138,139]. Fluid restriction and therapy with carbonic anhydrase inhibitors (e.g., topiramate, acetazolamide, zonisamide) compound the risk while on the diet. Renal calculi formation might be prevented by increasing fluid intake, prescription of oral citrates for the patients with a concomitant personal or family history of nephrolithiasis, discontinuation of ASMs which increase the risk of stone formation [21,140]. KD discontinuation due to nephrolithiasis is currently rare [23].

Linear growth may be impaired while on KDs. Retrospective clinical reviews indicate a height deceleration occurring over time while on the diet and this effect was independent of mean age, KD duration and calorie or protein intake per body weight [141,142]. A prospective study of 237 children treated with CKD found that younger children grew poorly in comparison with older children who grew “almost normally” [143]. The mechanism of growth retardation is largely unknown but may include influence of metabolic alterations on the growth hormone axis and osteopenia and/or osteoporosis with associated fracture risk [130,144,145]. A higher risk of decreased growth and bone fractures was reported in children maintained on the diet for more than 2 years [130].

Some side-effects were reported based on isolated case reports and their consistent relationship to the KD is unknown (e.g., cardiomyopathy, prolonged QT interval syndrome, acute pancreatitis, hepatic dysfunction, valproic acid hepatotoxicity, susceptibility to infection, hypoproteinemia) [21].

Quite recently, a systematic review of 45 prospective studies, including 7 randomized controlled trials regarding safety and tolerability of the KD used for the treatment of refractory childhood epilepsy was published. The most common adverse events included gastrointestinal disturbances (40.6%), hyperlipidemia (12.8%), hyperuricemia (4.4%), lethargy (4.1%), hypoproteinemia (3.8%), severe respiratory failure and pancreatitis occurred in less than 0.5% of children. The total retention rates of the diet for 1 and 2 years were 45.7% and 29.2 %, respectively [146]. Adverse effects were not the main reason for the KD discontinuation, nearly half of the patients discontinued the diet because of lack of efficacy [139]. In fact, authors observing patients on KD for prolonged periods of time (over 2 years) state that KD is very safe, with mild side-effects, growth retardation remaining the main problem that needs to be overcome [147]. Of great concern is KDs effects that could have implications for the patients’ long-term health, including atherosclerosis, osteoporosis, liver, and muscle dysfunction that need elucidation in the future.

## 7. Barriers to the Application of KD

Undoubtedly, KD is a currently well-established treatment for patients with medically refractory, nonsurgical epilepsy. Despite that fact, dietary therapy research is underfunded compared to drug research and unlike most medications, dietary therapy is not easily accessible worldwide. Moreover, other alternative treatments for patients with refractory epilepsy include vagus nerve stimulation, new generation anti-epileptic drugs, responsive focal cortical stimulation and corpus callosotomy [148,149]. Despite all barriers enlisted below, in the UK itself, KD patient numbers have increased from 101 in 2000 (22 centers) to 754 in 2017 (39 centers), which constitutes over 700% increase in 17 years [150].

To guarantee a possible treatment success with the use of KD, multiple elements need to be put together such as the patients’ or caregivers’ motivation; a skillful, experienced and well-organized keto-team and a fully supportive health care/insurance system (synergic task). However, management of these elements is unfortunately very demanding and time consuming, often leading to early treatment discontinuation or failure.

Patient—The most important element of treatment success is patient treatment adherence. KD therapy (it is good to call it metabolic therapy because word “diet” may seem trivial to some patients) is a very patient-centered type of treatment. Most important cause of poor retention is suboptimal efficacy, with the patient right to determine, if his/her efficacy of the diet is optimal or not [39]. Patients achieving their success are more likely to tolerate the diet even if it is somewhat unpalatable or difficult. Secondly, some patients complain about diet restrictiveness; however, less successful patients are more likely to call the diet restrictive. There are also culinary, ethnic, and social reasons—in some countries (e.g., China, Korea) food refusal is high because traditionally Asian diet is low in fat and heavily starched-based [151], some teenagers want to go out with peers and may feel isolated. A patient on the diet should understand that sticking to the treatment is his/her own responsibility and that even tiny amounts of cheating can spoil the overall effect of the diet. Common problems occur due to poor cooperation between keto-team and parents or caregivers. Very active parental involvement and effort is essential, which is not always achieved. This is due to lack of knowledge, lack of common goal, lack of access to appropriate foods, lack of support from family and financial reasons (too high costs of diet and visits due to lack of reimbursement and appropriate procedures in some countries). Low or medium resources countries tend to achieve much worse results after implementation of the diet [152] Poor tolerability and side-effects may limit the retention especially in less effective cases but may also cause cessation of the diet in severe complications such as pancreatitis and chronic vomiting [146].

Health care system—Lack of support from hospital authorities, reimbursement, and trained personnel are the main obstacles to overcome. In some countries physicians are equipped to apply a KD; however, the health insurance system does not recognize it as an important therapeutic method to justify reimbursement. In general, the poorer the country the more probable to encounter problems with the “non-supportive” system. Regarding the analysis of cost-effectiveness of the KD, Dutch authors state that KD fails to outweigh the cost of ASMs [153].

Keto-team (physician, dietician, psychologist)—Lack of knowledge and disagreement about the benefits and effectiveness of KD, sometimes pressure from peers to discontinue “bizarre therapy” or “off-label” use, lack of communication and cooperation in the team and awareness of performing time consuming activities without reimbursement are limiting success rate. Despite existing guidelines, academic meetings dedicated to KD and available high-quality scientific publications, many physicians still believe that the KD should be used as a last resort. Most important factors that increase awareness are foundations, namely: Charlie’s Foundation (USA) and Mathew’s Friends (UK) that managed worldwide popularization of the diet. Again, low- to medium-income countries are less likely to build up successful teams; however, good results are still possible with maximal engagement and motivation [94,122].

## 8. What We Still Need to Learn

Despite numerous studies exploring the mechanisms, efficacy, and side-effects of the KD, there is a lack of class I and II studies. There are mostly class III and IV studies available and further studies are required. That is why there is no agreement about the seizure types and syndromes that are likely to respond to the diet. Possible predictors of treatment success before or early on into the diet therapy have not been defined. Long-term use of the diet and the duration of its efficacy are not documented. There is no final agreement regarding the role of introductory fasting. The actual role of various types of fat to create a diet plan is not established. Furthermore, there is no agreement about the role of electroencephalographic (EEG) changes (with or without clinical improvement) in evaluating treatment success or failure. Even the very essential question, regarding whether ketogenic ratios matter for clinical efficacy seems to be unanswered. Although side-effects are not very serious when the diet is conducted carefully, some problems need to be overcome such as growth and height retardation. The question arises if we can avert low insulin level consequences regarding its influence on growth? Mechanisms operative in a given case are usually unknown. If we consider them as biochemical, hormonal, cellular, enhancement of physiological effects in a successful patient, are they singular? multiple? parallel? synergistic? or one predominant mechanism occurs?

Regarding the putative mechanisms, all theories concentrate on direct impact of diet on neurons. However, the question arises, if we should go beyond the brain. Maybe focusing on inflammatory mechanisms such as the role of kynurenines or inflammasomes [88,154] may yield significant step forward. Anaplerosis (replenishing TCA cycle intermediates, e.g., through triheptanoin supplementation) has been shown to favorably restore bioenergetics function and prevent ictogenesis in Glut1DS [155,156]. It is of note if these findings could be extended broadly beyond Glut1DS. Finally, very intriguing novel mechanism of KD seems to be related to the microbiome alterations following its application [97]. Thorough understanding of the mechanism of gut-brain axis relationship with respect to microbiome would help modify this mechanism enhancing KD effects. Apart from epilepsy, the preclinical data provides strong support of the efficacy of KD in a variety of diverse animal models of neuropsychiatric disorders. Although the evidence from clinical studies are encouraging, especially regarding Alzheimer’s disease, psychotic and autism spectrum disorders, in fact, it is very limited mainly to case reports and small pilot studies. Firm conclusion on the efficacy on KD cannot be drawn due to lack of randomized, controlled clinical trials [5,157].

These problems as well as many yet unasked questions warrant further investigation to enhance quickly growing field of metabolic therapy for epilepsy.

## Figures and Tables

**Figure 1 nutrients-12-02616-f001:**
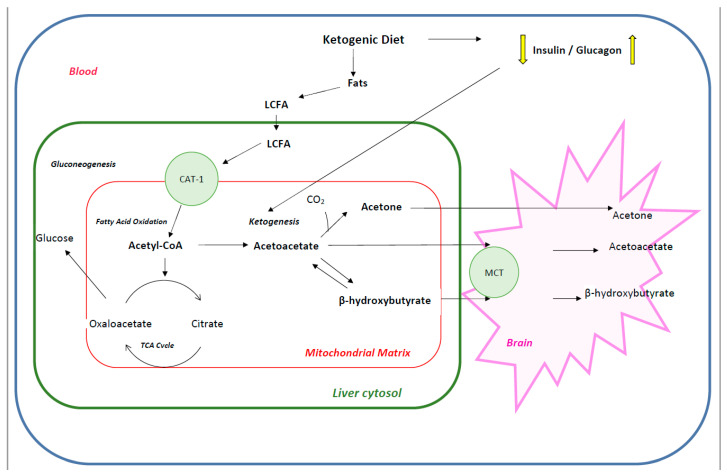
Depiction of liver cytosol metabolism that lead to ketone bodies formation penetrating to the brain (detailed description in the text). Abbreviations: CAT—carnitine-acylcarnitine translocase, LCFA—long-chain fatty acids, MCT—monocarboxylic acid transporter, TCA—tricarboxylic acid.

**Figure 2 nutrients-12-02616-f002:**
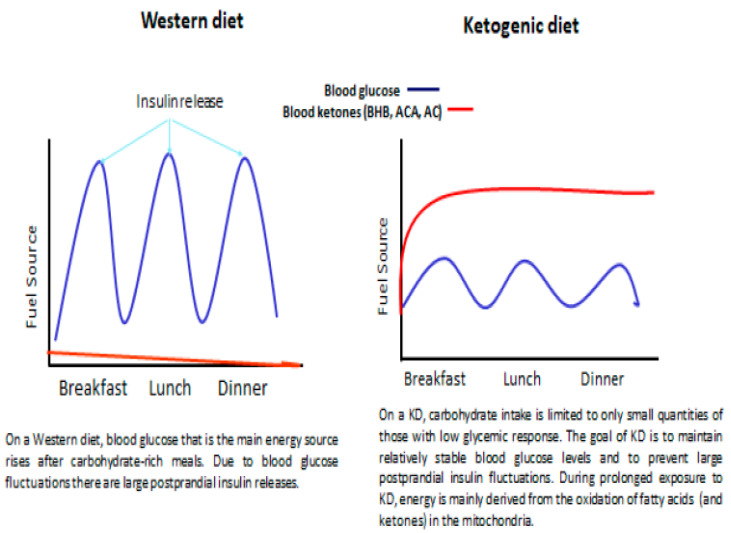
Differences in primary fuel sources between a typical Western diet and a KD. Visual depiction is taken from practice paper of the Academy of Nutrition and Dietetics [19], KD—Ketogenic diet.

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
