# Peer review of "Therapeutic Use of the Ketogenic Diet in Refractory Epilepsy: What We Know and What Still Needs to Be Learned"

_nutrients, 2020, doi:10.3390/nu12092616_

Round 1

Reviewer 1 Report

The manuscript is an intersting analysis of the published literature which can be useful for the readers, but there are some comments for the authors:

Introduction. In my opinion, this part is too long and should be shortened.

 Section 3. This section is really clear and well-written but more data about the low-glycemic index diet should be added. Please, read and cite the paper by Coppola g et al. Seizure 2011; 20:526-8

  • Section 6. A large part of this section should be re-written because there are many strong statements (e.g. Nowadays, KD...adulthood) that must be changed with softer claims.
  • Section 8. One of the most important barriers is the possible interference of KD on the plasma levels of common used antiepileptic drugs: this aspect must be discussed. Please, read and cite the paper by Coppola G et al. Acta Neurol Scand 2010; 122:303-7.
  • Section 9. In this section,  the real impact of KD on the arterial morphology must be discussed in detail: please, read and cite the paper by Coppola G et al. Seizure 2014; 23:260-5.
  •  

Author Response

Dear Reviewer,

Thank you for valuable comments.

Introduction. In my opinion, this part is too long and should be shortened.

 Introduction has been shortened (red track changes)

Section 3. This section is really clear and well-written but more data about the low-glycemic index diet should be added. Please, read and cite the paper by Coppola g et al. Seizure 2011; 20:526-8

Paragraph has been added on the results of LGIT in Italy

Section 6. A large part of this section should be re-written because there are many strong statements (e.g. Nowadays, KD...adulthood) that must be changed with softer claims.

The manuscript has been checked and I found two, three strong statements, these have been corrected

Section 8. One of the most important barriers is the possible interference of KD on the plasma levels of common used antiepileptic drugs: this aspect must be discussed. Please, read and cite the paper by Coppola G et al. Acta Neurol Scand 2010; 122:303-7.

Paragraph on the results of 2 authors has been added

Section 9. In this section,  the real impact of KD on the arterial morphology must be discussed in detail: please, read and cite the paper by Coppola G et al. Seizure 2014; 23:260-5.

Although the results of the authors are not conclusive, but rather contradictory, the results of 2 authors have been added

Best wishes,

Iwona Zarnowska

Reviewer 2 Report

This is a well written review but does not add much more to what has already been published in the field. Perhaps adding a discussion on the role of ketogenic diet and its many variants in other neurological diseases, besides epilepsy, will add more to the paper and make stand out from many other recent reviews.

Author Response

Thank you for the comments:

We felt that this lengthy manuscript should be on epilepsy only (title). But due to the reviewer remark a paragraph towards the end of the manuscript has been added.

Apart from epilepsy, the preclinical data provides strong support of the efficacy of KD in a variety of diverse animal models of neuropsychiatric disorders. Although, the evidence from clinical studies are encouraging, especially regarding Alzheimer’s disease, psychotic and autism spectrum disorders, in fact, it is very limited mainly to case reports and small pilot studies. Firm conclusion on the efficacy on KD cannot be drawn due to lack of randomised, controlled clinical trials [5, 157].

Best wishes,

Iwona Zarnowska

Round 2

Reviewer 1 Report

The manuscript is improved; no other changes are requested.